# Knock-Down of HDAC2 in Human Induced Pluripotent Stem Cell Derived Neurons Improves Neuronal Mitochondrial Dynamics, Neuronal Maturation and Reduces Amyloid Beta Peptides

**DOI:** 10.3390/ijms22052526

**Published:** 2021-03-03

**Authors:** Harald Frankowski, Fred Yeboah, Bonnie J. Berry, Chizuru Kinoshita, Michelle Lee, Kira Evitts, Joshua Davis, Yoshito Kinoshita, Richard S. Morrison, Jessica E. Young

**Affiliations:** 1Department of Laboratory Medicine and Pathology, University of Washington, Seattle, WA 98195, USA; haryf@uw.edu (H.F.); yeboafr1@uw.edu (F.Y.); bonniejb@uw.edu (B.J.B.); ckino@uw.edu (C.K.); mlee96@uw.edu (M.L.); kira.evitts@gmail.com (K.E.); jpd007@ucsd.edu (J.D.); ykino@uw.edu (Y.K.); 2Institute for Stem Cell and Regenerative Medicine, University of Washington, Seattle, WA 98109, USA; 3Molecular and Cellular Biology Graduate Program, University of Washington, Seattle, WA 98195, USA; 4Department of Neurological Surgery, University of Washington, Seattle, WA 98195, USA; yael@uw.edu

**Keywords:** hiPSCs, neuronal differentiation, histone deacetylase 2, endophilin-B1, mitochondria

## Abstract

Histone deacetylase 2 (HDAC2) is a major HDAC protein in the adult brain and has been shown to regulate many neuronal genes. The aberrant expression of HDAC2 and subsequent dysregulation of neuronal gene expression is implicated in neurodegeneration and brain aging. Human induced pluripotent stem cell-derived neurons (hiPSC-Ns) are widely used models for studying neurodegenerative disease mechanisms, but the role of HDAC2 in hiPSC-N differentiation and maturation has not been explored. In this study, we show that levels of HDAC2 progressively decrease as hiPSCs are differentiated towards neurons. This suppression of HDAC2 inversely corresponds to an increase in neuron-specific isoforms of Endophilin-B1, a multifunctional protein involved in mitochondrial dynamics. Expression of neuron-specific isoforms of Endophilin-B1 is accompanied by concomitant expression of a neuron-specific alternative splicing factor, *SRRM4*. Manipulation of HDAC2 and Endophilin-B1 using lentiviral approaches shows that the knock-down of HDAC2 or the overexpression of a neuron-specific Endophilin-B1 isoform promotes mitochondrial elongation and protects against cytotoxic stress in hiPSC-Ns, while HDAC2 knock-down specifically influences genes regulating mitochondrial dynamics and synaptogenesis. Furthermore, HDAC2 knock-down promotes enhanced mitochondrial respiration and reduces levels of neurotoxic amyloid beta peptides. Collectively, our study demonstrates a role for HDAC2 in hiPSC-neuronal differentiation, highlights neuron-specific isoforms of Endophilin-B1 as a marker of differentiating hiPSC-Ns and demonstrates that HDAC2 regulates key neuronal and mitochondrial pathways in hiPSC-Ns.

## 1. Introduction

Epigenetic dysregulation is a feature of many neurologic disorders, leading to aberrant gene expression that affects cellular metabolism and function [1]. Histone acetylation and deacetylation is a common epigenetic modification that can regulate gene expression by controlling chromatin structure. The extent of acetylation is regulated by balanced actions of histone acetyltransferases (HATs) and histone deacetylases (HDACs), regulates integral neuronal functions such as synaptic plasticity [2,3,4,5,6] and is important in neurodevelopment [7,8,9]. Imbalance of histone acetylation and deacetylation contributes to neuronal dysfunction in neurodegenerative disease [10,11].

In particular, histone deacetylase 2 (HDAC2) is an abundant HDAC in the brain and has been shown to specifically regulate genes involved in cognition, learning, and memory [4,5]. Dysregulation of HDAC2 has been implicated in neurodegenerative disorders including Alzheimer’s Disease (AD) [3,10,12], where it may contribute to cognitive impairment [5]. Knock-down of HDAC2 leads to up-regulation of synaptic gene expression in primary murine neurons [6]. Recently, we demonstrated that HDAC2 is involved in regulation of neuronal mitochondrial dynamics via the expression of Endophilin-B1 (*SH3GLB1*) in primary murine neurons. This work showed that increased HDAC2 sensitized neurons to mitochondrial dysfunction and cell death in neurodegenerative conditions [13].

Human induced pluripotent stem cells (hiPSC) are increasingly utilized for in vitro studies of neurologic disorders. Given that epigenetic dysregulation [14,15], mitochondrial dysfunction [16,17,18], and synaptic dysfunction [19] are all implicated in multiple neurodevelopmental and neurodegenerative diseases, we investigated whether modulation of HDAC2 expression would impact neuronal maturation, mitochondrial dynamics and synaptic gene expression in neuronal cells derived from hiPSCs (hiPSC-Ns). Here we show that HDAC2 levels progressively decrease during neuronal differentiation and this inversely correlates with a natural increase in neuron-specific isoforms of Endophilin-B1 (Endo-B1b/c). We demonstrate that small-hairpin RNA (shRNA)-mediated knock-down of HDAC2 promotes Endo-B1b/c expression in hiPSC-derived cortical neurons, confirming that the regulation of Endo-B1b/c by HDAC2 in human neurons. We then dissect the roles of HDAC2 and Endo-B1b/c in human neurons using knock-down and overexpression experiments to demonstrate that both knock-down of HDAC2 and overexpression of Endo-B1c promote mitochondrial elongation and protect neurons from cytotoxic stress but only knock-down of HDAC2 influences the expression of genes involved mitochondrial gene expression and mitochondrial respiration. Finally, we show that reduction in HDAC2 in hiPSC-Ns decreases levels of Amyloid beta (Aβ), the cleavage product of the amyloid precursor protein (APP) implicated in neurotoxicity in AD. Our data confirm, in a human neuronal cell model, the role of HDAC2 in modulating neuronal synaptic gene expression, and also implicates several pathways in which HDAC2 modulates mitochondrial dynamics and physiology in hiPSC-Ns. HDAC2 inhibition may represent a therapeutic strategy for AD and other neurodegenerative disorders.

## 2. Results

### 2.1. Expression of HDAC2 and Endophilin-B1 (Endo-B1) Isoforms in hiPSC-Derived Neurons

Our previous work demonstrates a role for HDAC2 in regulation of mitochondrial dynamics in primary mouse cortical neurons [13]. While class I HDACs, HDACs 1 and 2, are required for neuronal specification from neural progenitor cells [20] and control synapse function and maturation [8] in mice, HDAC2 expression and regulation of cellular processes in living, human neurons has not yet been examined. To begin to understand the role of HDAC2 in human neurons, we differentiated cortical neurons from well-characterized hiPSC lines [21,22,23,24] following our standard protocols [24,25,26] (Schematically depicted in Figure 1A). We harvested protein lysates from pluripotent stem cells (hiPSCs), neural progenitor cells that have not yet been directed to a neuronal lineage (NPCs), and neuronally differentiating NPC cultures at week 1, week 2 and week 3 time points. We observed that HDAC2 protein expression is present at all time points, with higher levels in neural progenitor cells that decline as neuronal differentiation proceeds (Figure 1B,C). Interestingly, endogenous expression of all class I HDACs (HDAC 1, 2, and 3) decreased with neuronal differentiation (Appendix A). This is consistent with work showing HDAC expression in neural progenitor cells in the developing mouse cortex [27]. Our previous work in mouse primary cortical neurons demonstrated that expression of HDAC2, but not other class I HDACs, negatively impacts the expression of Endophilin-B1 (Endo-B1), a multifunctional protein involved in mitochondrial dynamics [13]. In neurons, Endo-B1 is alternatively spliced yielding neuron-specific Endo-B1b and Endo-B1c as major isoforms relative to the ubiquitously expressed Endo-B1a [28]. Both of the neuron-specific isoforms are neuroprotective with Endo-B1c showing stronger activity in attenuating apoptotic cell death and causing mitochondrial elongation in mouse cortical neurons [28]. We therefore examined endogenous levels of Endo-B1 isoforms in neuronally differentiating cultures. As the cultures differentiate, we observed an increase in the neuron-specific isoforms of Endo-B1, which co-migrate on a Western blot (designated as Endo-B1b/c), compared to the ubiquitous isoform Endo-B1a, which migrates at a lower molecular weight (Figure 1B,D). We next examined the expression of HDAC2 and Endo-B1 isoforms specifically in differentiating neuronal cells compared with hiPSCs and NPCs. To this end, we enriched our cultures for neurons at each week of neuronal differentiation using magnetic bead sorting according to a modification of previously published protocols [29,30]. We observed that HDAC2 mRNA expression is dynamically controlled as hiPSCs differentiate toward neuronal lineages (Figure 1E). As the neuron-specific Endo-B1 protein isoforms cannot be separated on Western blot, we designed primers to specifically detect Endo-B1a, Endo-B1b, and Endo-B1c mRNA isoforms and observed strong increases in Endo-B1b and Endo-B1c mRNA during neuronal differentiation with little change observed for Endo-B1a (Figure 1F). The neural-specific splicing factor *SRRM4* has recently been implicated in alternative splicing of Endo-B1 pre-mRNA, favoring the generation of Endo-B1b and Endo-B1c isoforms over Endo-B1a [31]. We therefore analyzed *SRRM4* levels during neuronal differentiation and observed that endogenous *SRRM4* is highly upregulated at week 1 of neuronal differentiation (Figure 1G), corresponding to the appearance of Endo-B1b and Endo-B1c isoforms at this stage (Figure 1B,F). While *SRRM4* levels decline towards the end of the three-week neuronal differentiation, *SRRM4* mRNA levels are still significantly increased compared with expression in NPCs and hiPSCs (Figure 1G).

### 2.2. HDAC2 Knock-Down Influences Neuronal Gene Expression

Previous studies have implicated HDAC2 as a regulator of synaptic and cognitive gene expression [3] and shown that HDAC2 knock-down (HDAC KD) in mouse primary neurons leads to increases in synaptic gene expression [6]. As we observed a natural decrease in HDAC2 levels as our hiPSC-Ns differentiate and mature, we tested whether an experimentally evoked decrease in HDAC2 in differentiated hiPSC-Ns would further enhance expression of neuronal genes. In these experiments we differentiated neurons for three weeks and enriched by bead sorting. We transduced neurons with a lentivirus carrying a shRNA against HDAC2 and observed a strong decrease in *HDAC2* mRNA (Figure 2A). In these HDAC2 KD cells we observed a significant increase in *TBR1* (Figure 2B), a transcription factor expressed in deep layer cortical neurons [33], suggesting that KD of HDAC2 promotes cortical neuron identity in vitro. In concordance with previous studies [6], we observed a significant increase in the mRNA expression of key synaptic genes: *SYNGR3*, *PSD95*, *SHANK2* and *SHANK3* (Figure 2C–F), suggesting that HDAC2 is a repressive regulator of synaptic gene expression in human neurons as well.

### 2.3. HDAC2 Knock-Down Upregulates the Expression of Endo-B1b/c and SRRM4

The increasing Endo-B1b/c expression accompanied by declining HDAC2 expression during neuronal differentiation (Figure 1) suggests that the drop in HDAC2 may be promoting Endo-B1b/c expression. We next tested whether HDAC2 KD further increased neuronal isoforms of Endo-B1. We infected differentiated and enriched neurons with the HDAC2 shRNA lentivirus and observed a significant decrease in HDAC2 protein and increase in the Endo-B1b/c protein as compared to Endo-B1a (Figure 3A–C). We also observed a further increase in *SRRM4* mRNA (Figure 3D) and increase in the *Endo-B1b* and *Endo-B1c* mRNA splice isoforms (Figure 3E).

### 2.4. HDAC2 and Endo-B1c Expression Influences Mitochondrial Length

Impaired mitochondrial dynamics and function is a hallmark of neurodegeneration in multiple models [34,35]. Chemical pan-HDAC inhibitors have been shown to induce mitochondrial elongation [36]. Therefore, we first examined whether HDAC2 KD induces similar changes in mitochondrial shape in human neurons. We analyzed hiPSC-Ns stained with MitoTracker to measure mitochondrial length in the neurites and observed a significant increase in mitochondrial length in HDAC2 KD neurons compared to viral controls (Figure 4A,B). Modulation of HDAC2 changes neuronal mitofusin 2 (MFN2) and mitochondrial fission factor (MFF) expression in mouse primary neurons [13]. Consistently, we observed that while HDAC2 KD only resulted in a small increase in *MFN2* mRNA, it caused a marked decrease in *MFF* mRNA (Figure 4C,D). At the protein level, MFN2 protein levels were not different in HDAC2 KD neurons whereas MFF protein levels were significantly reduced (Figure 4E–G). Our results suggest that mitochondrial elongation apparent in HDAC2 KD neurons is mediated, at least in part, by a reduction in MFF levels.

Neuron-specific isoforms of Endo-B1 can also influence neuronal mitochondrial dynamics and previous work shows that Endo-B1c had the strongest effect on mitochondrial length in mouse neurons [28]. Therefore, we used a lentivirus to infect hiPSC-Ns with a construct that specifically overexpresses the Endo-B1c isoform and analyzed neuritic mitochondrial length. We document strong Endo-B1c overexpression (Figure 5A) and a significant increase in mitochondrial length in neurites (Figure 5B,C). However, overexpression of Endo-B1c did not significantly affect *MFF* or *MFN2* mRNA expression (Figure 5D,E). This is consistent with previous work showing no effect of Endo-B1 KD on these proteins [28] and suggests that Endo-B1b/c may affect fusion/fission protein activity in neuronal mitochondria rather than regulating gene expression of these fusion/fission proteins. Taken together, our results indicate that HDAC2 KD induces mitochondrial elongation in human neurons and suggest that it is mediated, at least in part, by increased Endo-B1b/c and decreased MFF expression as a result of HDAC2 KD.

### 2.5. HDAC2 Expression Influences Neuronal Viability and Mitochondrial Respiration

Mitochondrial dynamics in cells contributes to viability and metabolism [37]. Endo-B1c has been shown to be anti-apoptotic in neuronal cells while promoting mitochondrial elongation [28], so we hypothesized that HDAC2 KD would also be neuroprotective due to the increased expression of Endo-B1c. We knocked down HDAC2 or overexpressed Endo-B1c in hiPSC-Ns and challenged them with the cytotoxic agent camptothecin, a DNA topoisomerase I inhibitor that induces p53-dependent neuronal apoptosis [38,39]. We observed significant protection from camptothecin-induced cell death, as monitored based on caspase-3 activity, in both HDAC2 KD and Endo-B1c overexpression conditions (Figure 6), suggesting that HDAC2 KD can protect against neuronal cell death stimuli partly through elevated expression of neuron-specific Endo-B1 isoforms influencing mitochondrial elongation. Similarly, mitochondrial dynamics may influence respiration. Overexpression and knock-down of MFN2 has been shown to increase and decrease mitochondrial respiration, respectively [40] and in hiPSC-Ns knock-down of MFN2 has been shown to decrease mitochondrial bioenergetics [41]. Inhibition of fission can also impact mitochondrial energy production [42]. We measured oxygen consumption rate (OCR), which is an indicator of mitochondrial respiratory activity, in HDAC2 KD neurons using a Seahorse Bioscience XF96 analyzer. We observed a significant increase in basal and maximal OCR and in the spare respiratory capacity in HDAC2 KD neurons (Figure 7A–D). We next tested whether overexpression of Endo-B1c had an effect on mitochondrial respiration but found no significant difference in any of the OCR parameters (Figure 7E–G). An increase in mitochondrial biogenesis could explain the increase in respiration we observe in HDAC2 KD cells, however we did not detect a difference in mtDNA copy number in either HDAC2 knock-down neurons or Endo-B1c overexpressing neurons (Figure 7H). Together, these data suggest that decreasing HDAC2 levels in neurons may enhance mitochondrial respiration by regulating expression of genes that directly regulate metabolism.

### 2.6. HDAC2 Knock-Down Reduces Amyloid Beta (Aβ) 1–40 and 1–42 Peptides

Previous studies have demonstrated that HDAC2 expression is specifically increased in neurons in the AD brain using immunohistochemical approaches and HDAC inhibition may have therapeutic potential for neurodegenerative disease [3,43]. One of the neuropathological hallmarks of AD is senile plaques comprised of Aβ peptides generated by amyloidogenic cleavage of APP. Aβ peptides can be of various lengths with Aβ 1–40 being the most abundant and Aβ 1–42 being more neurotoxic [44]. We tested whether HDAC2 KD influenced Aβ peptides secreted by hiPSC-Ns in the culture media. hiPSC-Ns from AD patients as well as from non-demented controls show detectable levels of Aβ [23,24,29] and levels of these peptides can be reduced by treatment with various small molecules [24,45,46]. We infected wild-type (WT) hiPSC-Ns and hiPSC-Ns derived from a patient with a duplication of APP (APP^Dp^) [47], which has been used and characterized in other studies [29], with either the control or HDAC2 KD lentivirus. We measured the levels of secreted Aβ peptides in the culture media and observed that in both WT and APP^Dp^ hiPSC-Ns, HDAC2 KD reduced the levels of both Aβ 1–40 and Aβ 1–42 (Figure 8). We see a similar magnitude in Aβ peptide reduction from both the control and the APP^Dp^ cell lines, suggesting that HDAC2 does not impact this pathway differently based on the presence or absence of a familial AD mutation and thus may be beneficial in several treatment scenarios.

## 3. Discussion

Acetylation and deacetylation of histones is critical for regulating gene expression and is essential to normal neuronal development and function with dysregulated acetylation/deacetylation contributing to development of neurodegenerative conditions. Expression of the epigenetic regulator HDAC2 is altered in neurodegenerative diseases such as Alzheimer’s disease and this may influence expression of genes related to cognition [3]. Previously, we demonstrated that HDAC2 regulates expression of neuron-specific isoforms of Endo-B1, a protein that confers neuroprotection and promotes mitochondrial elongation, uncovering a novel role of HDAC2 in regulation of mitochondrial function [13]. In the current study, we sought to understand how HDAC2 expression influences hiPSC-N maturation and mitochondrial size and function. We first assessed HDAC2 expression during neuronal differentiation from hiPSCs and demonstrated progressively decreasing levels of HDAC2 mRNA and protein with time. HDAC2 is a class I HDAC that is important for neurodevelopment [48]. Studies in olfactory receptor neurons in the olfactory epithelium of mice showed that HDAC2 is highly expressed in early post-mitotic neurons, but not glia, and is downregulated during neuronal maturation [7,49]. Other studies show that HDAC2 expression is important for silencing neural progenitor transcripts during adult neurogenesis in the mouse [50]. Together, these studies indicate that HDAC2 expression is dynamic in differentiating and maturing neurons. In our study, we detect HDAC2 expression at mRNA and protein levels at all stages of the cells we studied (pluripotent-hiPSCs, neural progenitor cells-NPCs, and differentiating neurons). Interestingly, HDAC2 levels appear to increase one week after neuronal induction from NPCs and then decrease as neurons further differentiate and mature, however there is still measurable expression of HDAC2 in differentiated neurons on par with the level in hiPSCs. Future work looking at expression of class I HDACs in differentiating neurons and glial cells will be important in determining the complex roles these HDACs play in human neural development.

As we noted a decline in HDAC2 levels as hiPSC-Ns matured during neuronal differentiation, we hypothesized that the decline in HDAC2 levels may facilitate expression of genes that support neuronal function. Previous work in mouse primary neurons showed that knock-down of HDAC2 induces expression of synaptic genes by directly regulating histone acetylation at the promoter [6], which is complementary to data suggesting that upregulation of HDAC2 negatively affects expression of genes involved in cognition [3]. Consistent with these results, using an shRNA approach to further decrease HDAC2 levels in our differentiated hiPSC-Ns we revealed that decreased HDAC2 levels lead to increased expression of pre- and post-synaptic genes. Although our results cannot rule out an indirect effect of HDAC2 on genes we did not measure, these previous and current findings suggest that decreasing HDAC2 levels in neurons, including human neurons, may promote neuronal maturation and function and that lowering HDAC2 levels may be a strategy to restore normal function in neurologic disorders.

Previously we reported that HDAC2 regulates expression of Endo-B1, a multifunctional protein involved in mitochondrial dynamics. Specifically, we showed that decreasing HDAC2 levels in mouse neurons inversely elevates neuron-specific and neuroprotective Endo-B1 isoforms [13]. Neuron-specific isoforms of Endo-B1 promote neuronal survival and their expression in the brain is reduced in mouse models and human models of AD and mouse models of stroke [28,32]. We observed an increase in neuron-specific isoforms, Endo-B1b/c, as our human neurons differentiate and mature, consistent with the decreasing HDAC2 expression concurrently observed. Prompted by previous reports implicating the neuron-specific splicing factor *SRRM4* in the alternative splicing of Endo-B1 [31,51], we examined whether changes in *SRRM4* expression correlate with the appearance of Endo-B1b/c isoforms. We found that *SRRM4* mRNA expression is indeed induced upon neuronal differentiation and further demonstrated that *SRRM4* mRNA is upregulated by HDAC2 KD in differentiated neurons, suggesting that *SRRM4* gene expression may be negatively regulated by HDAC2 in human neurons. Thus, our data suggest that lowering HDAC2 levels may promote expression of *SRRM4* facilitating neuron-specific splicing of Endo-B1, which in turn can allow the resulting neuron-specific Endo-B1b/c isoforms to promote mitochondrial elongation. As expected, overexpression of Endo-B1c in hiPSC-Ns had a significant effect on mitochondrial elongation as well.

While knock-down of HDAC2 increases mitochondrial elongation likely through the induction in Endo-B1c, we also examined expression of MFN2 and MFF, molecules that promote fusion and fission of mitochondria, respectively. Our previous work showed that HDAC2 overexpression increases MFF expression while decreasing MFN2 expression in mouse neurons [13]. Consistently, HDAC2 KD in differentiated hiPSC-Ns conversely induced a significant decrease in *MFF* mRNA and protein although no significant effect was observed for *MFN2* mRNA. Separately, we confirmed that Endo-B1c overexpression has no effect on *MFF* or *MFN2* mRNA expression, ruling out any transcriptional activity of Endo-B1c, consistent with the reported cytosolic and/or mitochondrial localization of Endo-B1 [52]. Endo-B1c may thus have a more physical role in elongation of mitochondria. Indeed, recent work shows that Endo-B1 can regulate the mitochondrial inner membrane through an interaction with prohibitin-2 [53]. Taken together, our data suggest that modulation of HDAC2 in hiPSC-Ns influences mitochondrial dynamics, in part, through regulation of the expression of fusion and fission proteins, which includes alternative splicing-regulated production of the net fusion-promoting Endo-B1b/c isoforms (with the assistance of concomitantly regulated expression of *SRRM4*) and transcriptional regulation of *MFF* gene.

In addition to mitochondrial elongation, increased mitochondrial biogenesis is reported in terminally differentiated mouse cortical neurons [54]. In human pluripotent stem cell differentiations, cells undergo a shift from glycolysis in neural progenitors to oxidative phosphorylation in differentiating neurons [55]. We therefore decided to examine the effects of HDAC2 KD and the resulting upregulation of Endo-B1b/c on mitochondrial respiration and biogenesis in hiPSC-Ns. Analyses using a Seahorse Flux analyzer demonstrated that HDAC2 KD evokes a significant increase in OCRs representing the basal respiration, the maximal respiration and the spare respiratory capacity. This suggests that lowering levels of HDAC2 improves multiple aspects of mitochondrial physiology towards, for instance, more neuronally mature and thus more oxidative modes of metabolism and augmented bioenergetic capacity to confer increased resistance to stress. Interestingly, a similar action of HDACs has been reported in HL-1 cells derived from mouse atrial cardiac muscle, where HDAC inhibition with a class I HDAC-specific inhibitor, MPT0E014, improves mitochondrial OCR following TNF-alpha treatment to model heart failure [56]. These findings suggest that reducing HDACs may be a conserved process that could benefit mitochondrial bioenergetics across many different tissues. Interestingly, overexpression of Endo-B1c did not significantly affect these respiratory parameters. This indicates that the improved respiratory function in HDAC2 KD neurons is not solely a result of increased mitochondrial length. The absence of an increase in oxidative phosphorylation despite mitochondrial elongation induced by Endo-B1c overexpression suggests that these two functions are not always dependent on each other. Indeed, a carboxy-terminal truncation of MFN2 has been shown abrogate its mitochondrial fusion capacity but was still able to induce an increase in mitochondrial membrane potential and stimulate glucose oxidation [40]. Finally, we also tested whether the changes in mitochondrial respiration in HDAC2 KD cells were due to increases in mitochondrial biogenesis, however we did not see significant differences in mtDNA copy number in either HDAC2 KD or Endo-B1c OE cells.

As decreased HDAC2 levels induce Endo-B1c expression, which is known to be anti-apoptotic in neurons, we tested whether HDAC2 KD or Endo-B1c overexpression would protect hiPSC-Ns against a neurotoxic insult. Both HDAC2 KD and Endo-B1c overexpression significantly reduced caspase-3 activity induced by camptothecin treatment. These data suggest that lowering HDAC2 levels, either by genetic or pharmacologic means, may be a viable strategy in maintaining or restoring viability of human neurons during disease conditions and that forced expression of Endo-B1c could substitute such HDAC2 manipulation.

Mitochondrial and synaptic dysfunction are implicated in neurodegenerative disease pathogenesis and strategies to improve these pathways can be considered for novel therapeutic development. Indeed, HDAC inhibition has been suggested as a possible therapeutic pathways for neurodegeneration [43]. We tested whether HDAC2 knock-down could improve a cellular phenotype related to AD pathology, Aβ secretion into neuronal culture media. In both control and APP^Dp^ neurons, knock-down of HDAC2 significantly reduced the levels of secreted Aβ 1–40 and Aβ1–42 peptides. These data suggest that further investigation into HDAC inhibition, specifically HDAC2 inhibition, as a therapeutic strategy for AD is warranted.

In this study, we report a role for endogenous HDAC2 during human neuronal differentiation. Our data suggest a repressive regulation of the neuron-specific splicing factor *SRRM4* by HDAC2 and highlights Endo-B1b/c isoforms as novel functional contributors to and markers of human cortical neuronal differentiation. Using lentiviral knock-down and overexpression approaches we further confirm previous work in mice demonstrating that HDAC2 acts repressively on synaptic gene expression and we solidify the finding that HDAC2 regulates genes involved in mitochondrial bioenergetics and dynamics by showing, for the first time, this effect in hiPSC-Ns. Finally, we show a significant effect of HDAC2 on neuronal mitochondrial respiration and neuroprotection. Our data support the idea that manipulation of HDAC2 may be beneficial in the treatment of neurological diseases.

## 4. Materials and Methods

### 4.1. Cell Culture and hiPSC Neuronal Differentiation

These experiments represent data obtained from two well characterized control hiPSCs lines (CV and WTC11) [57,58].

The data showing reduction in Aβ in FAD cells was obtained from a cell line with an APP duplication [29].

The CV line was generated at the University of California, San Diego and was transferred to Dr. Young via a Material Transfer Agreement. The APP^Dp^ cell lines was generated at the University of California, San Diego and was transferred to Dr. Young via a Material Transfer Agreement. The WTC11 line was obtained from Dr. Carol Ware at the University of Washington’s Ellison Stem Cell Core. hiPSCs were cultured under feeder-free conditions and differentiated to neural progenitor cells (NPCs) following published protocols utilizing dual SMAD inhibition and after 12 days NPCs were purified using sorting for CD184+/CD24+ populations as previously described [30,59,60]. Neuronal differentiation of NPCs to cortical neurons was performed as we have previously published [23,25,26]. Briefly, NPCs were seeded at 10 × 10^6^ per 10 cm plate and differentiated for three weeks in the presence of GDNF (20 ng/mL PeproTech, Rocky Hill, NJ, USA), BDNF (20 ng/mL PeproTech, Rocky Hill, NJ, USA) and dbcAMP (250 ug/mL Sigma, St. Louis, MO, USA). To analyze cultures enriched in neurons, differentiated cultures were dissociated into a single cell suspension using Accutase (Innovative Cell Technologies, San Diego, CA, USA) and incubated with antibodies against CD184-PE and CD44-PE (BD Biosciences, San Jose CA, USA). Anti-PE magnetic beads (BD Biosciences, San Jose CA, USA) were added, and complexes were pulled down. The neuronally enriched supernatant was re-plated in a modification of the sorting protocol published by Yuan et al. [30] and used in our previous publications [24,25,26].

### 4.2. Lentivirus Treatment

Lentiviruses carrying a control shRNA, HDAC2 shRNA, GFP or Endo-B1c plasmids were generated as previously described [3,13]. For all lentivirus treatments, neurons were differentiated for three weeks and enriched by bead sorting. Four days after enrichment, viruses were added to the cultures and left for three days. Cells were further maintained in virus-free medium for 2 weeks and then harvested or used for assays.

### 4.3. RNA Purification and qPCR Analysis

Total RNA was purified from 2 × 10^5^ cells using TRIzol (Life Technologies, Carlsbad, CA, USA) followed by 1st strand cDNA synthetized using the iScript kit (Biorad, Hercules, CA, USA). Between 5 and 10 ng of cDNA were used in a 4 uL reaction using POWEUP SYBR qPCR mix (Life Technologies, Carlsbad, CA, USA). All primer-sets were run in technical triplicates. Expression was calculated using the 2—∆∆Ct method and genes of interest were normalized to RPL27. qPCR primers designed over exon-exon boundaries using Primer-Blast are below:
hHDAC2TGAGATTCCCAATGAGTTGCCATACTGACATCTGGTCAGACAhMFN2CACCCTGATGCAGACGGAAATCCATGTACTCGGGCTCTGAhMFFCAGCTTCACTAAGACGACAGATAATTACCTCTAGCGGCGAAACChPSD95CTCAGGGTCAACGACAGCATAAGCCAAGACCTTTAGGCCChRPL27TGAGATTCCCAATGAGTTGCCATACTGACATCTGGTCAGACAhSH3GLB1 isoform cCATGTAAAATGGCTGAAGATTTGGTGGGCATGTGTACTGCTGAThSHANK2CTGGCGAGCTGGGGTGATTATCAATGGGTGTGTCAGCTTTGhSHANK3CCTCACCTCACACAGCGATTCCACCGACTCGAGATACTGChSYNGR3CGTCCTGGGTGTTCTCCATCCTGCTGATTTGCTGGAAGCGhSRMM4ATAGCCCATCGCCTGTCAAGGCCGGCTTCGAGATTGTTTC

### 4.4. Western Blot Analysis

2 × 10^5^ cells were lysed in 50 μL RIPA buffer and protein amount was assessed using BCA assay kit (Thermo Scientific, Waltham, MA, USA). Between 3 and 10 μg of total protein were loaded on to a 4–15% gradient TGX gel (Bio-Rad) and transferred on to a PVDF membrane (Bio-Rad), which was then treated with 0.4% paraformaldehyde for 30 min to fix proteins. Following blocking with 5% nonfat dry milk, 0.1% Tween 20 and 0.05% thimerosal, membrane was incubated with the primary antibody diluted in 5% bovine serum albumin, 0.1% Tween 20, 0.05% thimerosal and 0.2% NaN_3_ overnight at 4 °C, followed by horse radish peroxidase-conjugated secondary antibody (1:2000, GE Healthcare, Chicago IL, USA) diluted in the blocking buffer. Membrane was then developed using Clarity Western ECL substrate (Biorad, Hercules, CA, USA) and exposed to Hyperfilm ECL (GE Healthcare, Chicago IL, USA). Films were digitally scanned, and band intensity was quantitated using ImageJ and normalized against β-actin.

Antibodies
HDAC2Sigma H2663 (2 mg/mL)1:5000Bif-1/Endophilin-B1(30A882.1.1)Novus #NBP2-24733 (0.5 mg/mL)1:1000MFN2Abcam #ab124773 (1.549 mg/mL)1:2000MFFProteintech #17090-1-AP (45 μg/150 μL)1:2000ActinSigma A54411:2500

### 4.5. Mitochondrial Length Analysis

Mitochondria were stained with MitoTracker Red (M7513 Invitrogen/ Thermo Scientific, Waltham, MA, USA) following manufacturer’s instructions, and pictures of at least three independent fields per sample were taken on a NIKON A1R confocal system using a 60x objective. Mitochondrial length was quantified by a blinded observer using ImageJ as we have previously described [28].

### 4.6. Caspase 3 Analysis

Neurons were cultured in a 96 well plate, in replicates, at a concentration of 2 × 10^5^ cells/well, treated with lentiviruses as indicated above (4.2) and were then treated for 24 h with 20 µM camptothecin or with vehicle (DMSO) alone. Cells were then lysed, and a caspase-3 assay was performed according to the manufacturer’s protocol (EnzChek caspase-3 Assay, Invitrogen/ Thermo Scientific, Waltham, MA, USA). The intensity of fluorescence was analyzed using an EnVision plate reader (PerkinElmer, Waltham, MA, USA).

### 4.7. Seahorse Analysis

Neurons were plated in a Matrigel-coated 96 well Seahorse plate at a density of 2 × 10^5^ cells/well and transduced with lentivirus as described above. The MitoStress protocol in the Seahorse XF96 Flux Analyzer (Agilent Technologies, Santa Clara, CA, USA) was performed two weeks later. An hour before the assay, the culture media was replaced with base media (Agilent Seahorse XF base medium, 103334-100 Agilent Technologies, Santa Clara, CA, USA) supplemented with 25 mM glucose and 1 mM Sodium pyruvate (11360070 Gibco/Thermo Scientific, Waltham, MA, USA ). Substrates and select inhibitors of the different complexes were injected during the measurement to achieve final concentrations of oligomycin (2.5 μM), FCCP (1 μM), rotenone (2.5 μM) and antimycin (2.5 μM). The oxygen consumption rate (OCR) values were then normalized with readings from Hoechst staining (HO33342 Sigma-Aldrich, St. Louis, MO, USA), which corresponded to the number of cells in the well.

### 4.8. Mitochondrial DNA Copy Number Analysis

Genomic DNA was prepared from samples in TRIzol (Life Technologies Carlsbad, CA, USA) that had previously been used to extract RNA according to manufacturer’s protocol. Then, 10 ng of genomic DNA were used per qPCR reaction and each sample was assessed in triplicate for mitochondrial ND1 (F: CCCTAAAACCCGCCACATCT, R: GAGCGATGGTGAGAGCTAAGGT) and nuclear LPL (F: CGAGTCGTCTTTCTCCTGATGAT, R: TTCTGGATTCCAATGCTTCGA) genes. Nuclear DNA-normalized, relative mitochondrial DNA content is provided by 2 × 2ΔCT where ΔCT = (nucDNA CT–mtDNA CT).

### 4.9. Aβ 1–40 and 1–42 Analysis

Aβ peptides were measured as previously described [24]. Briefly neurons were differentiated as described above in 4.1 and plated at 200,000 cells per well. The neurons were treated with lentiviruses as described above in 4.2. Medium was harvested to measure secreted Aβ peptides, media was run on an Aβ V-plex ELISA plate (Meso Scale Discovery, Rockville, MD, USA) per manufacturer’s instructions.

### 4.10. Statistical Analysis

Data represent two individual cell lines performed in biological and technical replicates. All data were analyzed using GraphPad Prism software v. 8 (GraphPad Software, Inc., La Jolla, CA, USA). For each data set, data were analyzed for normal distribution using the Shapiro–Wilk Test. Normally distributed data were analyzed using parametric statistical tests. For comparisons of more than two groups one-way ANOVA analysis was used with Tukey’s post hoc multiple comparisons test. For data in two groups two-tailed *t*-Tests were used. For non-normally distributed data, non-parametric tests were used. For non-normal data comparing two groups, the Mann–Whitney U test was used. Definition of replicates and all statistical tests used, and *p*-values are reported in the figure legends.

## Figures and Tables

**Figure 1 ijms-22-02526-f001:**
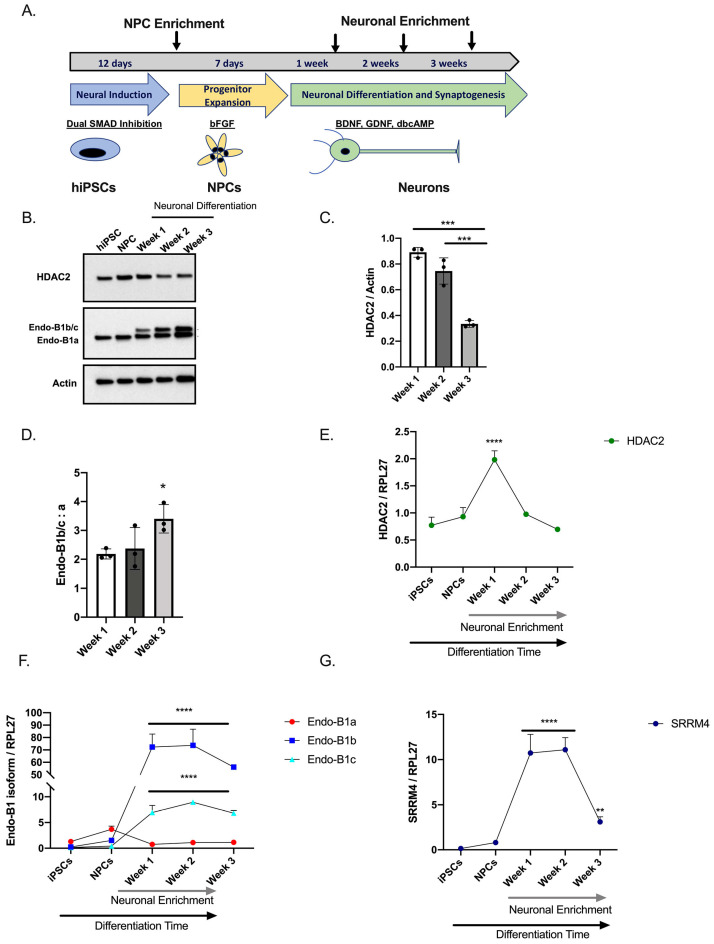
HDAC2 expression is dynamic during neuronal differentiation from hiPSCs and correlates with neuron- specific isoforms of Endophilin-B1, a protein that influences mitochondrial dynamics. (**A**). Schematic of neuronal differentiation protocol from hiPSCs. Arrows indicate time points when NPCs or neurons were enriched by cell sorting. (**B**). Representative Western blot analysis of endogenous HDAC2 and Endophilin-B1 levels in hiPSCs, NPCs, and during neuronal differentiation. Endo-B1a = ubiquitous isoform, Endo-B1b/c = neuron-specific isoforms. Note: Endo-B1b and Endo-B1c cannot be resolved on Western blot [13,28,32] and are indicated as Endo-B1b/c. (**C**). Quantification of HDAC2 protein decreases during a three-week neuronal differentiation (*N* = 3, *** *p* < 0.001 by one-way ANOVA with Tukey’s multiple comparisons test). (**D**). Quantification of the ratio of Endo-B1b/c to Endo-B1a protein isoforms during neuronal differentiation (*N* = 3, * *p* < 0.05 by one-way ANOVA with Tukey’s multiple comparisons test). (**E**). Endogenous *HDAC2* mRNA harvested from hiPSCs, NPCs, and neuronally enriched cultures harvested at 1, 2, and 3 weeks of differentiation. *HDAC2* mRNA expression increases during the first week of neuronal differentiation and then decreases substantially as neurons mature (*N* = 3, **** *p* < 0.0001 by one-way ANOVA with Tukey’s multiple comparisons test). (**F**). Quantification of *Endo-B1a*, *Endo-B1b*, *and Endo-B1c* mRNA isoforms in hiPSCs, NPCs, and neuronally enriched cultures harvested at 1, 2, and 3 weeks of differentiation. *Endo-B1b* and *Endo-B1c* mRNA expression is significantly elevated during neuronal differentiation relative to the levels in hiPSCs and NPCs while little change was observed for *Endo-B1a* mRNA (*N* = 3, **** *p* < 0.0001 by one-way ANOVA with Tukey’s multiple comparisons test) (**G**). Quantification of *SRRM4* mRNA in hiPSCs, NPCs, and neuronally enriched cultures harvested at 1, 2, and 3 weeks of differentiation. *SRRM4* is highly upregulated during the first two weeks of neuronal differentiation. While levels fall during the third week, *SRRM4* transcripts are still at significantly higher levels than in hiPSCs or NPCs (*N* = 3, ** *p* < 0.01, **** *p* < 0.0001 by one-way ANOVA with Tukey’s multiple comparisons test).

**Figure 2 ijms-22-02526-f002:**
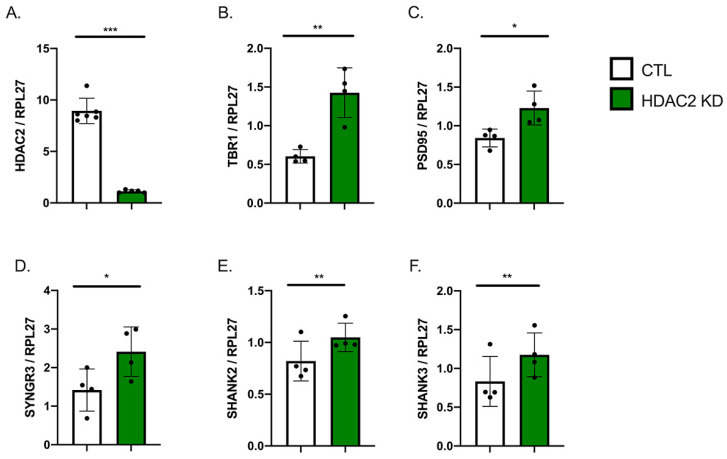
HDAC2 KD in hiPSC-derived neurons influences neuronal and synaptic gene expression. (**A**). Lentiviral transduction in hiPSC-Ns with an shRNA to HDAC2 significantly reduces HDAC2 mRNA. (**B**). Knock-down of HDAC2 (HDAC2 KD) increases mRNA of *TBR1*, a gene that influences cortical neuron identity. (**C**–**F**). HDAC2 KD increases expression of synaptic genes *PSD95* (**C**), *SYNGR3* (**D**), *SHANK2* (**E**), and *SHANK3* (**F**). HDAC2 KD vs. CTL neurons (Each dot represents the mean of 4–6 independent experiments, *** *p* <0.001 by *t*-test, ** *p* < 0.01 by *t*-test, * *p* < 0.05 by *t*-test).

**Figure 3 ijms-22-02526-f003:**
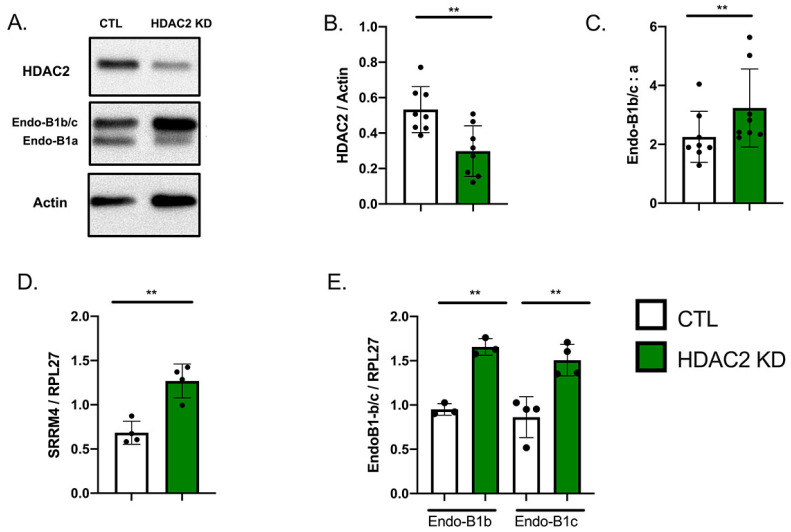
HDAC2 knock-down via shRNA further induces neuronal isoforms of Endo-B1 in hiPSC-derived neurons. (**A**). Representative Western blot demonstrating HDAC2 KD after lentiviral infection and resulting induction in Endo-B1b/c isoforms at the protein level. (**B**,**C**). Quantification of HDAC2 protein and Endo-B1b/c:a isoform ratio in 3-week differentiated neurons transduced with HDAC2 shRNA (*N* = 8, ** *p* < 0.01 by *t*-Test). (**D**). HDAC2 KD induces *SRRM4* mRNA expression. (**E**). HDAC2 KD induces *Endo-B1b* and *Endo-B1c* mRNA expression (Each dot represents the mean of 3–4 independent experiments, ** *p* < 0.01 by *t*-test).

**Figure 4 ijms-22-02526-f004:**
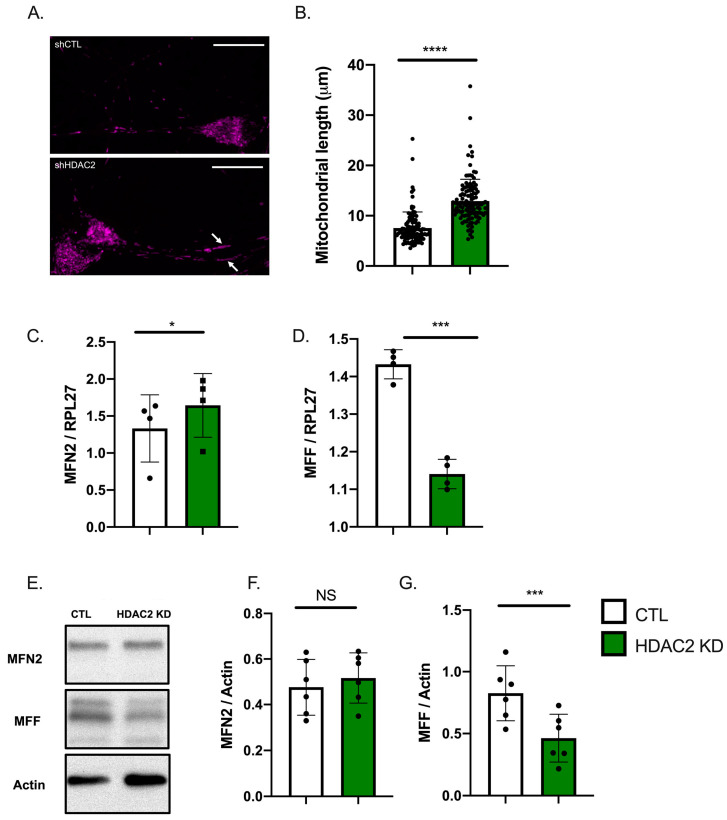
Knock-down of HDAC2 in hiPSC-derived neurons leads to elongated mitochondria in neurites and influences expression of genes involved in mitochondrial dynamics. (**A**). Representative image of elongated mitochondria visualized with Mitotracker in hiPSC-derived neurons with HDAC2 KD vs. CTL shRNA. (**B**). Quantification of mitochondrial length in neurites (*N* = 4 independent Mitotracker experiments, 59 control mitochondria and 57 HDAC2 KD mitochondria, **** *p* < 0.0001 by Mann–Whitney U test). (**C**). qRT-PCR analysis of Mitofusin 2 (*MFN2*) expression in HDAC2 KD (Each dot represents the mean of 4 independent experiments, * *p* < 0.05 by *t*-test). (**D**). qRT-PCR analysis of Mitochondrial Fission Factor (*MFF*) expression in HDAC2 KD (Each dot represents the mean of 4 independent experiments, *** *p* < 0.001 by paired *t*-test). (**E**). Representative Western blot analysis of MFN2 and MFF protein expression in CTL vs. HDAC2 KD conditions. (**F**,**G**). Quantification of MFN2 and MFF protein levels: MFN2 protein levels are unchanged in HDAC2 KD (*N* = 6, NS = not significant by *t*-test). MFF protein levels are decreased in HDAC2 KD (*N* = 6, *** *p* < 0.001 by *t*-test). Scale bar = 20 μm.

**Figure 5 ijms-22-02526-f005:**
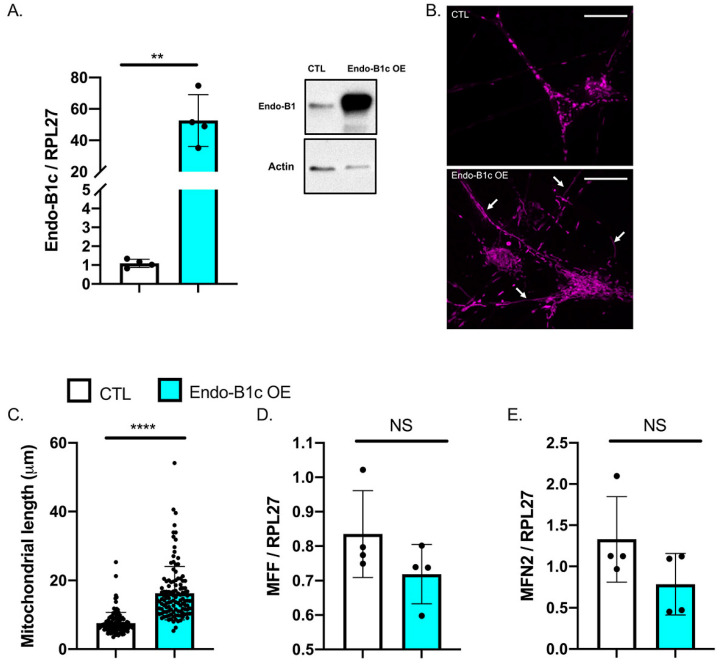
Overexpression (OE) of Endophilin-B1c in hiPSC-derived neurons leads to elongated mitochondria but does not influence *MFN2/MFF* gene expression. (**A**). mRNA quantification of Endo-B1c OE compared to a control OE vector (Each dot represents the mean of 4 independent experiments, ** *p* < 0.01 by *t*-test). Representative Western blot confirms overexpression of Endo-B1c protein. (**B**). Representative image of elongated mitochondria visualized with MitoTracker in hiPSC-derived neurons with Endo-B1c OE vs. a control overexpression vector. (**C**). Quantification of mitochondrial length in neurites (*N* = 4 independent MitoTracker experiments, 59 control mitochondria, and 62 Endo-B1c OE mitochondria, **** *p* < 0.0001 by Mann–Whitney U test). (**D**,**E**). qRT-PCR analysis of *MFF* (**D**) and *MFN2* (**E**) expression in Endo-B1c overexpressing neurons (Each dot represents the mean of 4 independent experiments, NS = not significant by *t*-test). Scale bar = 20 μm.

**Figure 6 ijms-22-02526-f006:**
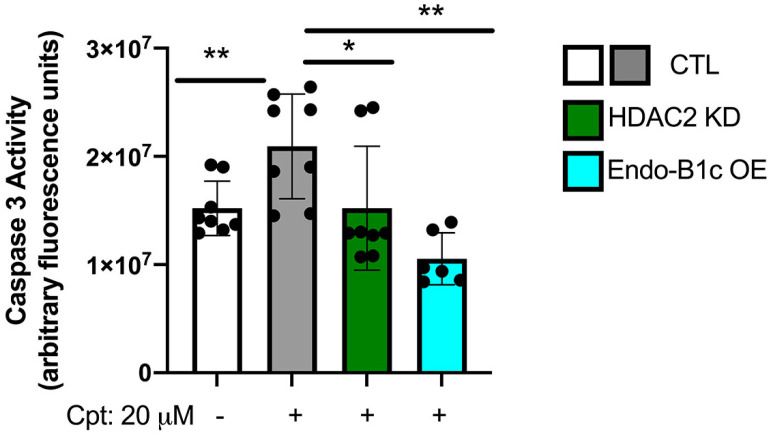
HDAC2 KD and OE of Endo-B1c in hiPSC-derived neurons is protective against neuronal stress. hiPSC-derived neurons were treated for 24 h with 20 μM camptothecin. Cell death was quantified by measuring caspase-3 activity. Each dot represents a technical replicate of 3–4 independent experiments. Analysis compares all groups to the CTL/untreated bar (white) using a one-way ANOVA with Tukey post hoc multiple comparisons; ** *p* < 0.01; * *p* < 0.05.

**Figure 7 ijms-22-02526-f007:**
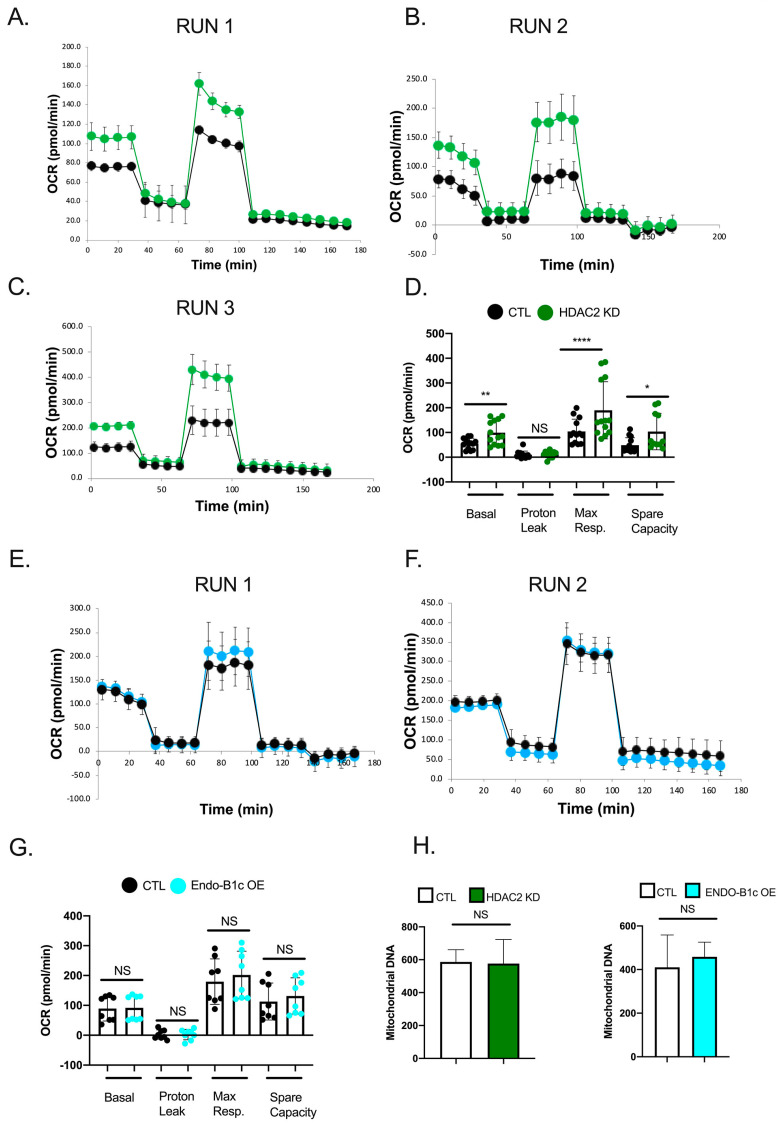
HDAC2 KD but not OE of EndophilinB1-c improves mitochondrial respiration in hiPSC-derived neurons. (**A**–**C**). Plots of individual Seahorse assays measuring the oxygen consumption rate (OCR) of hiPSC-Ns with HDAC2 KD compared to a control virus. (**D**). Compilation of the three independent Seahorse assays (shown in **A**–**C**) shows significant increases in basal respiration, maximum respiration (Max resp.), and spare capacity. Each dot represents a technical replicate within each independent assay and data are analyzed using a one-way ANOVA with Tukey post hoc multiple comparisons; * *p* < 0.05; ** *p* < 0.01; **** *p* < 0.0001. (**E**,**F**). Plots of individual Seahorse assays of hiPSC-Ns with EndoB1-c overexpression compared to a control virus. (**G**). Compilation of the two independent Seahorse assays (shown in **E**,**F**) shows no significant changes in basal respiration, maximum respiration (Max resp.), proton leak, or spare capacity. Each dot represents a technical replicate within each independent assay and data are analyzed using a one-way ANOVA with Tukey post hoc multiple comparisons; NS = non-significant. (**H**). Analysis of mitochondrial DNA copy number in hiPSC-Ns with HDAC2 KD or Endo-B1c OE. No significant difference was found in mitochondrial copy number in either condition when compared to a control virus. NS = non-significant by *t*-test.

**Figure 8 ijms-22-02526-f008:**
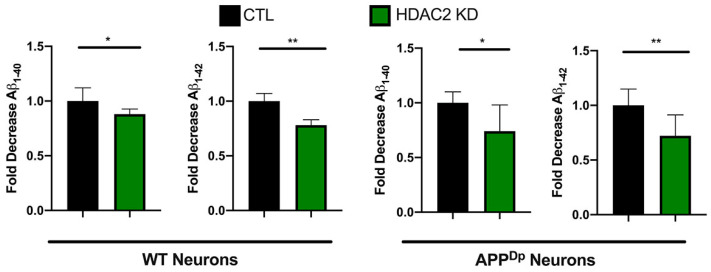
Knock-down of HDAC2 reduces Aβ peptides in hiPSC-Ns. Aβ peptides (1–40 and 1–42) secreted in the media from both WT and APP^Dp^ hiPSC-Ns were reduced under HDAC2 knock-down conditions. The data represent two independent differentiations, CTL vs. HDAC2 KD analyzed by *t*-test ** *p* < 0.01; * *p* < 0.05.

## Data Availability

Data is contained within the article. For additional requests please contact the corresponding author.

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
