# Peer review of "Knock-Down of HDAC2 in Human Induced Pluripotent Stem Cell Derived Neurons Improves Neuronal Mitochondrial Dynamics, Neuronal Maturation and Reduces Amyloid Beta Peptides"

_ijms, 2021, doi:10.3390/ijms22052526_

Round 1

Reviewer 1 Report

Journal: International Journal of Molecular Sciences

Manuscript ID: ijms-1095262

Title: Histone Deacetylase 2 (HDAC2) influences maturation and mitochondrial

dynamics in human induced pluripotent stem cell derived neurons.

Authors: Harald Frankowski, Fred Yeboah, Bonnie Berry, Chizuru Kinoshita,

Michelle Lee, Kira Evitts, Joshua Davis, Yoshito Kinoshita, Richard Morrison,

Jessica Young * Submitted to section: Molecular Neurobiology,

In this research, the authors extended their previous observations in mouse neuron cells to study the impact of HDAC2 on the maturation of human induced pluripotent stem cell-derived neurons. They observed that alteration in the HDAC2 level led to changes in mitochondria morphology. Overall, the research confirmed previous statement of the role of HDAC2 in neuronal maturation. However, the study is limited by lack of mechanistic studies, i.e., the lack of the subset of genes directly influenced by HDAC2 enzymatic activity. Moreover, the following suggestions are expected to improve the quality of the manuscript.

Major points:

Figure 1, have the authors checked the expression levels of the other HDACs such as HDAC1? Alternately, are there published results demonstrating HDAC2 play a dominant role in neuronal maturation?

Figures 2 and 3, Considering that HDAC2 is a histone deacetylase, it is suggested that the authors should check the status of the related region in the chromosome and transcriptional activity of the corresponding genes. These two figures only show that the steady state of the gene expression levels. Note that changes in steady state levels does not mean that HDAC2 modulated the transcription of these genes via histone modifications. The steady state level could be influenced by many factors.

Besides defects in mitochondrion, are there any other phenotypes observed in the knockdown cells?

Minor points:

Figure 1A, I can not find the reference 25 entitled “Leptomeninges-Derived Induced Pluripotent Stem Cells and Directly 552 Converted Neurons From Autopsy Cases With Varying Neuropathologic Backgrounds” in PUBMED. Has this figure demonstrating the mentioned standard protocol been published elsewhere? If so, the authors should clearly mention it in the figure legend and obtain copyright from the previous publication.

The legend of Figure 7 is confusing and needs to be reorganized.

The resolution of the figures looks quite low. It is expected to include high-resolution images in the manuscript.

The manuscript contains some typos. The authors should carefully proofread before re-submission. For example, line 18, what does “however” means? Line 22, “are is”? Line 75 “HDAC 1 and HDAC2” should be expressed as “HDAC1 and HDAC2” or “HDACs 1 and 2”.

Reviewer 2 Report

Dr. Harald Frankowski and colleagues performed studies with the use of human iPSC-based neuronal cell model that demonstrate a role of histone deacetylase 2 (HDAC2) in hiPSC-neuronal differentiation, as a regulator of key neuronal and mitochondrial pathways in neurons (hiPSC-Ns); and HDAC2 functions by repressing neuron-specific isoforms of Endophilin-B1b/c (Endo-B1b/c). In this study the authors convincingly showed that:  HDAC2 and Endo-B1b/c is dynamically expressed during hiPSCs neuronal differentiation. HDAC2 represses neuronal and synaptic gene expression in hiPSC-Ns. HDAC2 represses the expression of Endo-B1b/c and SRRM4 in hiPSC-Ns. Inactivation of HDAC2 and Endo-B1c over-expression increase mitochondrial length partly by HDAC2 activity to suppress expression of mitochondrial fission factor (MFF). HDAC2 suppresses neuronal viability and mitochondrial respiration in hiPSC-Ns. Endo-B1c enhances neuronal viability in hiPSC-Ns. The manuscript is very well written and organized, all points are strongly supported by experimental data and statistical analysis. There are few comments that needs to be addressed by the authors.

Comments:

1) In the title of the paper I would suggest to indicate an exact function of HDAC2 as a suppressor of maturation in neurons. For instance: “ Histone Deacetylase 2 (HDAC2) represses (or acting as a repressor in) maturation and mitochondrial dynamics in human induced pluripotent stem cell derived neurons”.

2) Due to use of one shRNA construct for each studied gene it is important to clarify that there is no off-target effects for these shRNA constructs. As seen in the previous study (reference 13, doi: 10.1111/bpa.12647) the authors used one of shRNA construct for each gene, and there is no reference indicating that at least 2 (better 3 shRNA constructs) showed the similar effect in gene silencing. If there is a reference showing that the used shRNA constructs were selected from 3 different shRNA constructs then it needs to indicate it in the text. If there was not performed such analysis before then it is important to show that 2 or 3 shRNA constructs of each gene shows the same effect in one particular analysis of their function and then the authors can use only one of these shRNAs in following experiments. 

3) In material and methods: line 383 – it is better to name – 4.1. Cell culture and human iPSC neuronal differentiation.

In 4.6. Caspase 3 analysis, it is important indicate – “Neurons were cultured and treated with lentiviruses as indicated above (4.2.).

4) There are some few misprints throughout the text: “are is” - line 22, “exidative modes” – line 345, “that that” - line 353.

Round 2

Reviewer 1 Report

The authors have addressed my previous concerns. While the results in Figure 8 is potentially interesting, this is just a preliminary investigation of the impact of HDAC2-KD on neurodegenarative diseases. Therefor, I suggest the authors to delete the phrase "and cellular neurodegenerative disease phenotypes " from the Title.

Author Response

We thank the reviewer for the feedback and we agree that the title is overstating the results. We have changed the title to:

Knock-down of HDAC2 in human induced pluripotent stem cell derived neurons improves neuronal mitochondrial dynamics, neuronal maturation and reduces Amyloid beta peptides. 

We hope this conveys the message of the paper without overstatement.